# Methodological transparency of preoperative clinical practice guidelines for elective surgery. Systematic review

Gustavo Angel[1], Cristian Trujillo[1], Mario Mallama[1], Pablo Alonso-Coello[2], Markus Klimek[3], Jose A. Calvache[1,3]*

1 Department of Anesthesiology, Universidad del Cauca, Cauca, Colombia, 2 Iberoamerican Cochrane Centre, Clinical Epidemiology and Public Health Department, Hospital de la Santa Creu i Sant Pau, Universitat Autonoma de Barcelona, Barcelona, Spain, 3 Department of Anesthesiology, Erasmus University Medical Centre Rotterdam, Rotterdam, The Netherlands

* jacalvache@unicauca.edu.co

## Abstract

### Background

Clinical practice guidelines (CPG) are statements that provide recommendations regarding the approach to different diseases and aim to increase quality while decreasing the risk of complications in health care. Numerous guidelines in the field of perioperative care have been published in the previous decade but their methodological quality and transparency are relatively unknown.

### Objective

To critically evaluate the transparency and methodological quality of published CPG in the preoperative assessment and management of adult patients undergoing elective surgery.

### Design

Systematic review and methodological appraisal study.

### Data sources

We searched for eligible CPG published in English or Spanish between January 1, 2010, and June 30, 2022, in Pubmed MEDLINE, TRIP Database, Embase, the Cochrane Library, as well as in representatives' medical societies of Anaesthesiology and developers of CPG.

### Eligibility criteria

CPG dedicated on preoperative fasting, cardiac assessment for non-cardiac surgery, and the use of routine preoperative tests were included. Methodological quality and transparency of CPG were assessed by 3 evaluators using the 6 domains of the AGREE-II tool.

**Data Availability Statement:** All relevant data are within the paper and its Supporting Information files.

**Funding:** This work was supported by departmental funding (Department of Anesthesiology, Universidad del Cauca, Colombia, and Department of Anesthesiology, Erasmus University Medical Centre Rotterdam, The Netherlands). The funders had no role in study design, data collection and analysis, decision to publish, or preparation of the manuscript.

**Competing interests:** The authors have declared that no competing interests exist.

## Results

We included 20 CPG of which 14 were classified as recommended guidelines. The domain of "applicability" scored the lowest (44%), while the domains "scope and objective" and "editorial interdependence" received the highest median scores of 93% and 97% respectively. The remaining domains received scores ranging from 44% to 84%. The top mean scored CPG in preoperative fasting was ASA 2017 (93%); among cardiac evaluation, CPG for non-cardiac surgery were CCS 2017 (91%), ESC-ESA 2014 (90%), and AHA-ACC 2014 (89%); in preoperative testing ICSI 2020 (97%).

## Conclusions

In the last ten years, most published CPG in the preoperative assessment or management of adult patients undergoing elective surgery focused on preoperative fasting, cardiac assessment for non-cardiac surgery, and use of routine preoperative tests, present moderate to high methodological quality and can be recommended for their use or adaptation. Applicability and stakeholder involvement domains must be improved in the development of future guidelines.

## Introduction

Much of the global burden of disease requires surgical intervention and over 234 million operations are conducted each year worldwide [1, 2]. Lack of timely access to high-quality surgical care remains a major problem in much of the world, even though surgical interventions can be cost-effective interventions in terms of lives saved and disabilities avoided [3]. In addition, perioperative period is frequently associated with morbidity and mortality [4]; in high-income countries, major complications are occurring in 3 to 16% of in-hospital surgeries leading to permanent disability or mortality ranging from 0.4 to 0.8%, while increased to 5% to 10% in low- and middle-income countries [5].

Clinical practice guidelines (CPG) are statements containing evidence-based recommendations aimed at providing better patient care and helping physicians and patients to make the best decisions for the prevention, diagnosis, and treatment of different diseases [6]. The use of CPG during preoperative assessment and treatment of patients before surgical procedures can contribute to reducing the risk of complications, increase patient safety, improve the quality of care, enable the implementation of effective interventions, decrease treatment variability, and finally improve patient outcomes [7].

The development of high-quality CPG is a complex, lengthy, and systematic scientific process involving developers, stakeholders, and users, and therefore, the development process and produced recommendations can present varying degrees of quality [6, 8]. CPG have been evaluated by several authors including numerous conditions and years of development and publication (from 1980 to 2019). In general, the applicability domain has received the lowest mean score, and improvement in time is still a matter of controversy [9–12].

On perioperative care, Barajas-Nava et al., evaluated 22 CPG from 1990 up to 2008 describing their quality as moderate for most of the domains with the lowest scores in stakeholder involvement, applicability and editorial independence [7]. Ciapponi et al. reported the domain "applicability" as the worst score in preoperative (2010 to 2017) CPG [13]. In addition, a recent evaluation of guidelines in airway management, as part of perioperative care, showed applicability scores of around 23% [14]. Studies were not able to detect any improvement over time in

CPG quality, specifically in this context [7, 14], and show opportunities to improve the quality of CPG development [13].

We aimed to critically evaluate the transparency and methodological quality of the development of published CPG in the preoperative assessment and management of adult patients undergoing elective surgery published in the last decade, by using the AGREE-II tool.

## Methods

### Design

Systematic review including published CPG in preoperative care with a descriptive, methodological, and quality appraisal approach [15]. The reporting of this systematic review was guided by the standards of the Preferred Reporting Items for Systematic Review and Meta-Analysis (PRISMA) Statement, table in S1 Appendix.

### Search strategy

We conducted an extensive search in PubMed MEDLINE, TRIP Database, Embase, and the Cochrane Library, as well as on 12 specific dedicated websites for CPG developers, text in S2 Appendix. In addition, we explored all relevant medical societies (World Federation of Societies of Anaesthesiologists (WFSA), American Society of Anesthesiologists (ASA), European Society of Anaesthesiology (ESA), Association of Anaesthetists of Great Britain and Ireland (AAGBI), and Canadian Anesthesiologists' Society (CAS)). Our search was limited to CPG published between January 1, 2010, and June 30, 2022, that were written or published in English or Spanish. All searches were performed in December 2021 and updated in June 2022. The protocol of this study was previously published (PROSPERO ID 200026) [16].

### Selection process

We defined a CPG as any document comprising clinical recommendations for the preoperative assessment or treatment of adult patients undergoing elective surgery classifying them into three categories related to 1) Preoperative fasting, 2) Cardiac assessment for non-cardiac surgery, and 3) Use of routine preoperative tests (laboratory test, X-ray, pulmonary function test, and electrocardiogram). CPG designed for the assessment and management of specific conditions or designed for specific individuals were excluded (i.e., obstructive sleep apnoea, diabetes mellitus, specific surgeries, or designed only for obstetric patients, paediatric patients, or other specific populations).

### Data collection process and quality appraisal

Two reviewers (C.T. and G.A) screened independently the records based on the eligibility criteria and three independent reviewers (C.T., G.A., and M.M.) conducted data extraction and assessment by using a validated form in the online tool My Agree Plus (https://www.agreetrust.org/). All evaluators underwent a thorough detailed training for the AGREE-II tool application by following two major online-training modules to assist users in effectively applying the tool. The first one contains an avatar-guided overview of the AGREE II tool and follows a step-by-step process to complete each item and domain. It also provides immediate feedback on how the trainees' responses compare with those of expert ratings. The second one is an online-based tutorial with a virtual coach accompanied by a practice appraisal exercise. This strategy has been previously tested [17].

Any discrepancy was solved by consensus with the advice of a fourth reviewer (J.A.C.). The following data were obtained: the main category of the CPG, the number of authors, year of

publication or update, type of institution (governmental institution, specialty society or consortium, and university), version of the guideline (first and revision/updated), region of origin, reported funding (yes/no), the method for guideline development (systematic review, consensus or narrative review, adaptation or adoption, and not mentioned), methods to formulate recommendations (formal, informal consensus, and not mentioned), and methods to grade evidence (yes/no).

We used the revised version of the Appraisal of Guidelines for Research and Evaluation (AGREE-II) to assess the quality of CPG. The AGREE is a validated and widely used tool to assess CPG quality, and its components are based on the elements for high-quality CPG defined by the National Academy of Medicine (formerly Institute of Medicine) and by the Guidelines International Network [8]. AGREE-II comprises 23 items organized in six domains and two global rating items (overall assessments). Each domain assesses a different dimension of the CPG quality: scope and purpose (domain 1), stakeholder involvement (domain 2), rigor of development (domain 3), clarity of presentation (domain 4), applicability (domain 5), and editorial independence (domain 6) (8). Each item was rated using a seven-point Likert scale ranging from 1 (strongly disagree) to 7 (strongly agree).

## Data analysis

The score for each AGREE-II domain was calculated as the sum of all scores of the individual items in the domain and the total was standardized as a percentage of the maximum possible score for that domain, using the following formula: (score obtained–minimum possible score)/(maximum score–minimum possible score) x 100. With the results from each evaluator, a summary table was designed to generate median values for each domain and interquartile range (IQR). Another researcher (J.A.C.) analysed results to obtain the degree of concordance of the evaluation; in which 'score obtained' was the sum of the scores by individual evaluators, maximum score = 7 (strongly agree) x 3 (evaluators) x number of items in the domain and minimum score = 1 (strongly disagree) x 3 (evaluators) x number of items in the domain [14, 18].

The overall mean quality score classified the CPG as 'recommended' ($>$60%), 'recommended with modifications' (30 to 60%), or 'not recommended' ($<$30%) to be applied in clinical practice. Finally, an absolute agreement among the three reviewers was determined by using the intraclass correlation coefficient with its 95% confidence interval, based on a mean-rating (k = 3), two-way random-effects model. A standardized score was calculated separately for each of the six domains, and it was classified as a poor agreement ($<$0.50), moderate (0.50 to 0.75), good (0.75 to 0.90), and excellent ($>$0.90). The data were analysed with the IBM SPSS 25.0 package for Windows (Armonk, NY: IBM Corp.), and R Statistics.

## Results

We included 20 CPG in our analysis (Fig 1). During our eligibility process, we excluded 14 CPG, text in S3 Appendix. Based on the mean quality score, 14 CPG (70%) were classified as "recommended" to be implemented in clinical practice (Tables 1 and 2). A total of 16 CPG (80%) were produced by medical societies, 12 (60%) were new guidelines (first version), and only 10 (50%) explicitly reported the source of funding for development. Most CPG (95%) described they used a systematic review process and only six (30%) reported a formal method for achieving consensus.

Considering all CPG, the highest AGREE-II median scores were observed in the domains "scope and objective" (93%), "clarity of presentation" (84%), and "editorial independence" (97%). The lowest median scores were assigned to "stakeholder involvement" and "applicability" (56% and 44%, respectively) (Fig 2, Table 2). Overall, CPG focused on the use of routine

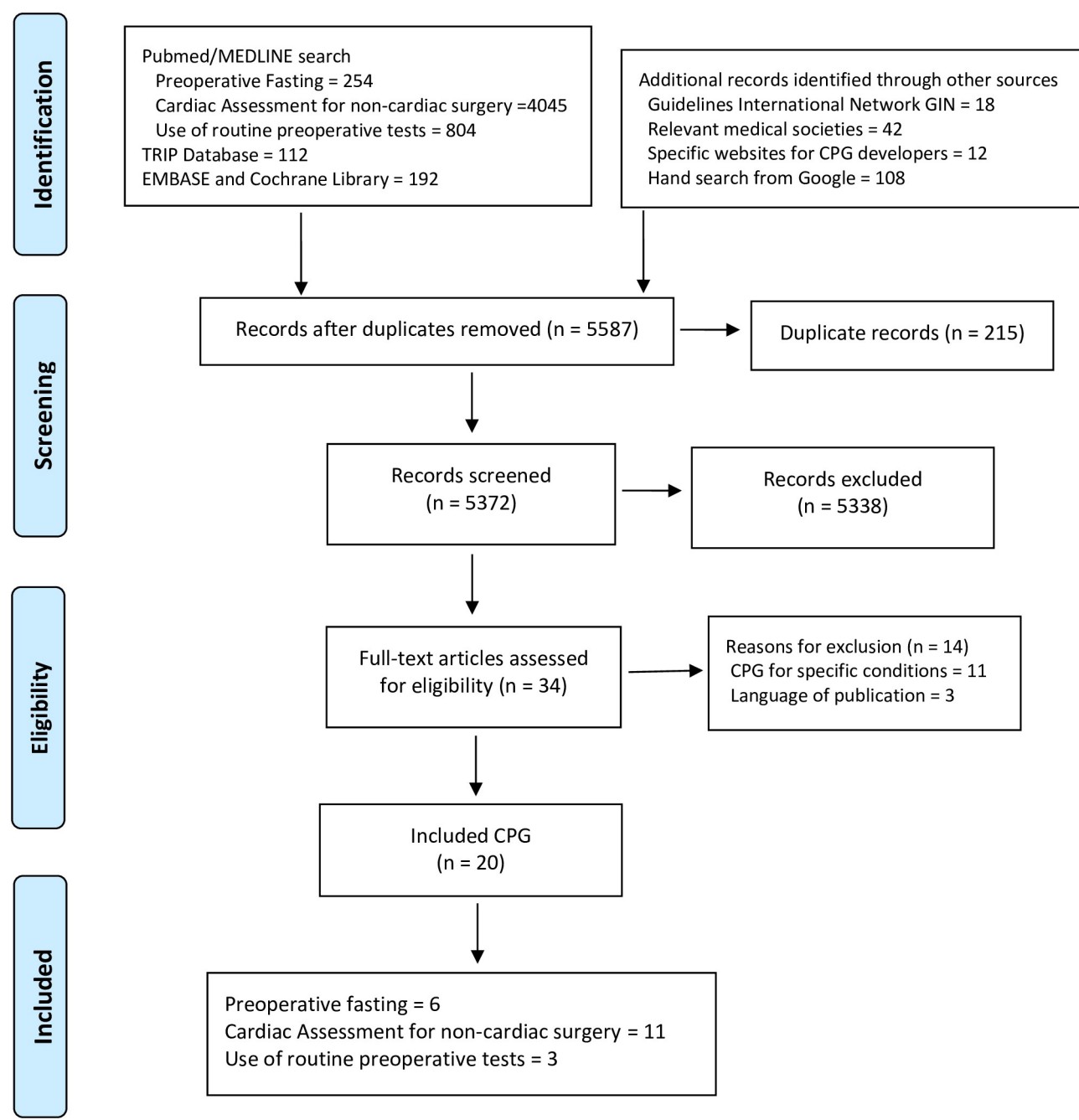

**Fig 1. Search and selection process of CPG.**

preoperative tests were rated higher than CPG related to cardiac assessment for non-cardiac surgery and CPG related to preoperative fasting (81% versus 73% and 74% respectively).

The highest mean rated CPG among the preoperative fasting category was the ASA 2017 (93%) [19]; among the cardiac evaluation for non-cardiac surgery CPG were CCS 2017 (91%) [20], ESC-ESA 2014 (90%) [21], and AHA-ACC 2014 (89%) [22]. Finally, ICSI 2020 [23] was the best-rated CPG in the preoperative testing category (97%) (Table 2). Inter-rater agreement was rated as good to excellent for all domains (Table 3).

**Table 1. Characteristics of included CPG stratified by recommendation status\* (n = 20).**

| Characteristic | No. (% of the total) | Recommended CPG n = 14 No. (% per category) | Recommended with modifications CPG n = 6 No. (% per category) |
|---|---|---|---|
| Main topic | | | |
| Preoperative fasting | 6 (30) | 4 (67) | 2 (33) |
| Cardiac assessment for non-cardiac surgery | 11 (55) | 8 (73) | 3 (27) |
| Use of routine preoperative tests | 3 (15) | 2 (67) | 1 (33) |
| Year of publication | | | |
| 2010 | 1 (5) | 1 (100) | 0 (0) |
| 2011 | 5 (25) | 3 (60) | 2 (40) |
| 2014 | 3 (15) | 2 (67) | 1 (33) |
| 2016 | 3 (15) | 1 (33) | 2 (67) |
| 2017 | 4 (20) | 4 (100) | 0 (0) |
| 2018 | 1 (5) | 1 (100) | 0 (0) |
| 2019 | 1 (5) | 0 (0) | 1 (100) |
| 2020 | 1 (5) | 1 (100) | 0 (0) |
| 2021 | 1 (5) | 1 (100) | 0 (0) |
| Number of authors | | | |
| ≤ 5 | 1 (5) | 0 (0) | 1 (100) |
| 6–10 | 5 (25) | 4 (80) | 1 (20) |
| 11–20 | 6 (30) | 4 (67) | 2 (33) |
| ≥ 20 | 8 (40) | 6 (75) | 2 (25) |
| Type of institution | | | |
| Governmental institution | 2 (10) | 2 (100) | 0 (0) |
| Specialty society or consortium | 16 (80) | 10 (63) | 6 (37) |
| University or academic institution | 2 (10) | 2 (100) | 0 (0) |
| Region | | | |
| United States | 4 (20) | 4 (100) | 0 (0) |
| United Kingdom | 2 (10) | 2 (100) | 0 (0) |
| Belgium | 2 (10) | 2 (100) | 0 (0) |
| Argentine | 2 (10) | 0 (0) | 2 (100) |
| Brazil | 2 (10) | 1 (50) | 1 (50) |
| Canada | 1 (5) | 1 (100) | 0 (0) |
| Germany | 1 (5) | 1 (100) | 0 (0) |
| Denmark | 1 (5) | 1 (100) | 0 (0) |
| Spain | 1 (5) | 0 (0) | 1 (100) |
| Japan | 1 (5) | 0 (0) | 1 (100) |
| Mexico | 1 (5) | 1 (100) | 0 (0) |
| Singapore | 1 (5) | 0 (0) | 1 (100) |
| South Africa | 1 (5) | 1 (100) | 0 (0) |
| Guideline version | | | |
| First version | 12 (60) | 9 (75) | 3 (25) |
| Revision or updated version | 8 (40) | 5 (63) | 3 (37) |
| Method for guideline development | | | |
| Systematic review | 19 (95) | 14 (73) | 5 (27) |
| Not mentioned | 1 (5) | 0 (0) | 1 (100) |
| Recommendation methods | | | |
| Not mentioned | 4 (20) | 0 (0) | 4 (100) |
| Informal consensus | 10 (50) | 9 (90) | 1 (10) |
| Formal consensus | 6 (30) | 5 (83) | 1 (17) |
| Funding | | | |
| Clearly reported | 10 (50) | 10 (100) | 0 (0) |
| Not mentioned | 10 (50) | 0 (0) | 6 (100) |

\* The overall quality AGREE-II score classified the CPG as 'recommended' (>60%), 'recommended with modifications' (30 to 60%), or 'not recommended' (<30%)

**Table 2. Scores of the AGREE-II domains for 20 included CPG stratified by category and recommendation status[*].**

| Category | CPG | AGREE-II domains | | | | | | |
|---|---|---|---|---|---|---|---|---|
| | | Scope and objective (%) | Stakeholder involvement (%) | Rigour of development (%) | Clarity of presentation (%) | Applicability (%) | Editorial independence (%) | Mean score (%) |
| Preoperative fasting | ASA 2017 [19] | 98 | 89 | 85 | 94 | 90 | 100 | **93** |
| | ESN 2017 [24] | 98 | 52 | 83 | 83 | 68 | 100 | **81** |
| | ESA 2011 [25] | 91 | 61 | 67 | 83 | 32 | 100 | **72** |
| | ASA 2011 [26] | 96 | 65 | 78 | 78 | 56 | 0 | **62** |
| | AAA 2016 [27] | 70 | 20 | 34 | 59 | 6 | 89 | **46** |
| | CAS 2019 [28] | 72 | 28 | 25 | 54 | 10 | 33 | **37** |
| Cardiac assessment for non-cardiac surgery | CCS 2017 [20] | 91 | 93 | 90 | 96 | 75 | 100 | **91** |
| | ESC-ESA 2014 [21] | 98 | 83 | 76 | 98 | 86 | 100 | **90** |
| | AHA-ACC 2014 [22] | 93 | 85 | 89 | 91 | 78 | 100 | **89** |
| | ESA 2018 [4] | 98 | 69 | 90 | 89 | 72 | 83 | **84** |
| | ESA 2011 [29] | 98 | 70 | 77 | 83 | 36 | 100 | **78** |
| | GFM 2010 [30] | 100 | 56 | 67 | 74 | 40 | 97 | **72** |
| | BSC 2017 [31] | 85 | 39 | 34 | 91 | 43 | 94 | **64** |
| | SA 2021 [32] | 72 | 50 | 56 | 78 | 39 | 83 | **63** |
| | SAC 2016 [33] | 81 | 30 | 64 | 87 | 38 | 0 | **50** |
| | BSC 2011 [34] | 65 | 28 | 25 | 85 | 38 | 53 | **49** |
| | JCS 2011 [35] | 57 | 54 | 41 | 65 | 44 | 0 | **44** |
| Use of routine preoperative tests | ICSI 2020 [23] | 94 | 98 | 95 | 98 | 97 | 100 | **97** |
| | NICE 2016 [36] | 98 | 56 | 61 | 93 | 81 | 100 | **81** |
| | SEA 2014 [37] | 93 | 52 | 31 | 65 | 3 | 97 | **57** |
| **Overall results** | **Median score** | **93** | **56** | **67** | **84** | **44** | **97** | |
| | **IQR** | **79 to 98** | **47 to 74** | **39 to 84** | **77 to 91** | **37 to 76** | **75 to 100** | |

[*]Recommended CPG classification (Grey shading, n = 14); Recommended with modifications (White shading, n = 6).

Mean AGREE-II scores among the included CPG ranged from 37% to 97% without major changes in the mean CPG scores over time (Table 2). Among the recommended CPG (n = 14, 62% to 97%), 10 reported funding, 2 were produced by governmental institutions, 10 by medical societies, and 2 by universities or academic institutions. In contrast, none of the CPG recommended with modifications reported funding, none were developed by governmental or academic institutions, and all were produced by medical societies. Overall, the most frequent characteristics presented in the recommended CPGs were being produced by governmental institutions (2 of 2, 100%), involving more than 20 authors (6 of 8, 75%), being the first version (9 of 12, 75%), reporting funding (10 of 10, 100%), and being developed by following a systematic review process (14 of 19, 73%) (Table 1). The number of included guidelines in this study prevents a more detailed analysis of associated factors.

## Discussion

This systematic review included 20 CPG related to preoperative assessment and management of adult patients undergoing elective surgery. In the cardiac evaluation for non-cardiac surgery guidelines, two CPG (CCS 2017 [20] and ESC-ESA 2014 [21]) obtained AGREE-II scores equal or above 90%; In preoperative fasting, one CPG (ASA 2017 [19]) obtained a score of

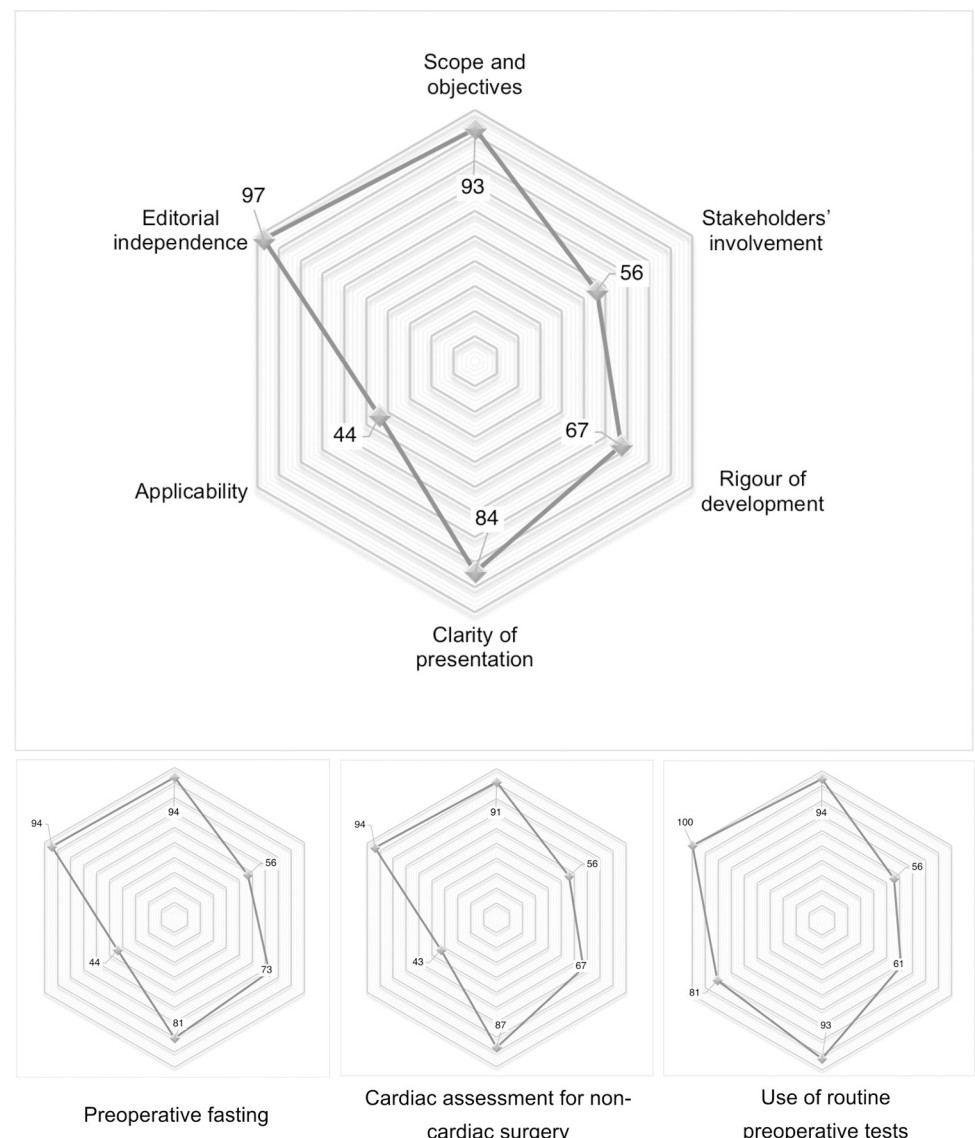

**Fig 2. Median scores of CPG evaluated in six domains of the AGREE-II instrument stratified by category (n = 20).**

**Table 3. Inter-rater agreement of AGREE II domains.**

| Domain | Inter-rater agreement* | | |
|---|---|---|---|
| | Intra-class correlation coefficient (ICC) | 95% CI | Degree of agreement |
| Scope and objective | 0.92 | 0.84 to 0.96 | Excellent |
| Stakeholder involvement | 0.96 | 0.92 to 0.98 | Excellent |
| Rigour of development | 0.96 | 0.92 to 0.98 | Excellent |
| Clarity of presentation | 0.86 | 0.59 to 0.94 | Good |
| Applicability | 0.95 | 0.90 to 0.98 | Excellent |
| Editorial independence | 0.96 | 0.90 to 0.98 | Excellent |

*Intra-class correlation coefficient. Poor agreement ($<0.50$), moderate (0.50 to 0.75), good (0.75 to 0.90), and excellent ($>0.90$).

93%, and in preoperative testing, one CPG (ICSI 2020 [23]) obtained a score of 97%. In general, 14 of 20 included CPG were classified as recommended to be used in clinical practice.

The domain "applicability" had the lowest scores; this has also been found extensively in other related studies [9–12, 38]. On perioperative care and airway management, applicability has been rated low [7, 13, 14]. Considering CPG are designed to be widely used, these findings represent important challenges to the guideline's development and implementation. Many of the current CPG do not consider the inclusion of tools and strategies that facilitate their application in real clinical scenarios or barriers that limit their use. Sometimes, CPG do not provide insights on how the recommendations can be put into practice or regarding the resource consequences of applying them (i.e., in limited resources settings). That can be translated into a low score in this domain. In addition, the opinion of patients or patients' representatives is rarely described in the development of recommendations resulting in low scores found in the domain "stakeholder involvement".

Recent developments of CPG may be reflected by a continuous increase in some domains [39]. The domain "scope and objective" received a high score, showing that most included guidelines have a clear objective, a well-defined population under consideration, and some specific focused clinical questions. In the perioperative scenario, this remains a high-rated domain without major differences from previous assessments [7, 13].

Editorial independence is critical in the process of developing high-quality CPG [40] and it has been previously reported as one of the lowest scored domains [7]. Results of this study show that in the preoperative assessment and treatment of adult patients before elective surgery, most available guidelines of the last decade can be considered independent, in line with a recent assessment [13]. Across domains, editorial independence obtained the highest score (97%) but there are some CPG scoring zero in this domain. While this finding adds confidence to the provided recommendations by reducing the risk of delivering biased recommendations or being influenced by sponsors (seeking their benefits) or by the pharmaceutical industry, there are still some CPG produced without a clear and transparent editorial independence. This large scoring variability also reflects the relative importance each sponsor gives to the editorial independence process during development, and the great implications to reviewers and evaluators during judging this domain. Most of the included CPG were produced by specialty societies but specific conflicts of interest were declared as well as their potential effect on the development process and content of the guideline with transparency. All CPG classified as recommended reported their funding sources while none of the recommended with modifications did.

Methodological procedures used to generate recommendations are critical steps outlined by AGREE-II as the "rigour of development" domain. In comparisons to previous studies [9] and other related perioperative scenarios [14], included guidelines presented a median score of 67%, with higher scores for CPG produced in Europe, United States, and Canada. From 1980 to 2007, a very slow increase in quality in CPG was reported [9] However, in the last decade, the AGREE-II tool has become the most widely used international "gold standard" for guidelines development [41], potentially supporting the improvement in this area.

Recently Ciapponi et al., published a systematic review focused to summarize and compare recommendations presented in evidence-based CPG for preoperative care [13]. Including 16 CPG, they reported many strong recommendations ready to be considered for implementation. In line with these findings, our results classified most included CPG as "recommended". In addition, when comparing our findings, they do not differ to a greater extent, being consistent that the domains top-scored were "scope and objective", "Clarity of presentation", and "editorial independence". Also, "applicability" was the lowest rated.

Low applicability can be explained by the difference in the populations to which the intervention is directed or by the lack of technological resources to perform such intervention in

different settings. In addition, as stated by Ciapponi et al., an inadequate adoption of guidelines (due to the tendency of many physicians to practice a "defensive medicine"), results in unnecessary interventions only to avoid legal problems and concerns derived from the care [13]. This is especially important for the use of routine preoperative tests.

There are three major barriers to CPG implementation described in the literature. Personal factors (related to physicians' knowledge and attitudes), guideline-related factors, and external factors [42]. In addition, central elements of successful strategies for CPG implementation include dissemination, education and training, social interaction, decision support systems, and standing orders. A whole adjusted and adapted process is needed in advance when surgical teams are trying to implement a new guideline. Recent evidence indicates that only by following a structured process that includes an analysis of the potential local-related barriers, implementation and adherence can be effective [42].

This study did have some limitations. First, AGREE-II items do not include a category "does not apply". In some cases, information to evaluate a single domain is missing and doubt remains as to how this should be scored and rated. Second, the degree of training required to allow an adequate and valid evaluation of a CPG is always a matter of concern, and certain subjective judgments may be present between evaluators. In our case, a detailed training process was implemented for the three evaluators regarding the application of the AGREE II tool, resulting in an agreement rated from good to excellent for all included areas. Third, to include updated versions of certain CPG may be controversial but it also allows us to compare the change in score with the previous versions. Finally, while most CPG are published in English, we included only CPG published in English and Spanish losing some relevant information from particular countries.

Healthcare providers use CPG considering guidelines to provide recommendations based on the best available evidence [43]. However, a 2009 study reported that most of the evidence used in the American College of Cardiology / American Heart Association CPG was based on expert recommendations [44], commonly non scientifically validated opinions. A systematic review of the evidence-based CPG published by the American and European scientific societies in anaesthesiology has recently been published. Their authors evaluated the quality of the evidence used to provide recommendations, finding that only 16% of them were based on level A of evidence, 33% on level B of evidence, and 51% on level C of evidence. These findings imply a critical need for greater efforts to improve the evidence used in the future CPG [43]. In our study, the limited number of CPG prevents a more detailed analysis of the factors associated with high methodological quality; further studies to assess the relationship between characteristics of CPG developers and the quality of CPG are warranted as they are available in other scenarios [45].

In 2022, O'Shaughnessy et al., assessed the quality of CPG published during the last 5 years in top anesthesia journals. With a scope beyond the preoperative care, they included 51 CPG with low scores to "stakeholder involvement", "rigor of development" and "applicability" domains [46] Additionally, Mai et al. in 2021, evaluated 96 CPG in anesthesiology practice finding "rigor of development" and "applicability" as the lowest rated domains [47]. Publication of a CPG in a peer-reviewed high-quality journal may enhances the scientific credentials of the process [48] while reducing the potential inclusion CPG produced by low- and middle-income countries or in any language different from English. In addition, time barriers, peer review and the overall editorial process may distort and delay the original message and recommendations as part of the development of the guideline [49, 50].

In conclusion, CPG in the preoperative assessment or management of adult patients undergoing elective surgery including preoperative fasting, cardiac assessment for non-cardiac surgery, and use of routine preoperative tests present moderate to high methodological quality

and can be recommended for their use or adaptation. Domains of applicability and stakeholder involvement must be improved in the development of future guidelines.

## Supporting information

**S1 File.**
(DOCX)

**S1 Appendix. PRISMA 2020- checklist of items that should be included in systematic review.**
(DOCX)

**S2 Appendix. Search strategies.**
(DOCX)

**S3 Appendix. Excluded CPG during eligibility process.**
(DOCX)

## Acknowledgments

**Assistance with the study:** The authors are grateful to the Medical Library, Erasmus University Medical Centre Rotterdam for assistance with the literature search.

**Presentation:** Preliminary data for this study were presented as a poster presentation at the Colombian Society of Anesthesiology meeting, 5–7 August 2021.

## Author Contributions

**Conceptualization:** Gustavo Angel, Cristian Trujillo, Mario Mallama, Pablo Alonso-Coello, Markus Klimek, Jose A. Calvache.

**Data curation:** Gustavo Angel, Cristian Trujillo, Mario Mallama, Markus Klimek, Jose A. Calvache.

**Formal analysis:** Jose A. Calvache.

**Funding acquisition:** Gustavo Angel.

**Investigation:** Gustavo Angel, Cristian Trujillo, Mario Mallama, Markus Klimek.

**Methodology:** Gustavo Angel, Cristian Trujillo, Mario Mallama, Pablo Alonso-Coello, Markus Klimek, Jose A. Calvache.

**Supervision:** Markus Klimek, Jose A. Calvache.

**Writing – original draft:** Gustavo Angel, Cristian Trujillo, Mario Mallama, Pablo Alonso-Coello, Markus Klimek, Jose A. Calvache.

**Writing – review & editing:** Gustavo Angel, Cristian Trujillo, Mario Mallama, Pablo Alonso-Coello, Markus Klimek, Jose A. Calvache.

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
