## [Decision Letter · Decision Letter 0]

8 Nov 2022

PONE-D-22-20952METHODOLOGICAL TRANSPARENCY OF PREOPERATIVE CLINICAL PRACTICE GUIDELINES FOR ELECTIVE SURGERY. SYSTEMATIC REVIEW.PLOS ONE

Dear Dr. Jose Andres Calvache,

Thank you for submitting your manuscript to PLOS ONE. After careful consideration, we feel that it has merit but does not fully meet PLOS ONE’s publication criteria as it currently stands. Therefore, we invite you to submit a revised version of the manuscript that addresses the points raised during the review process.

We look forward to receiving your revised manuscript.

Kind regards,

Seung-Hwa Lee

Academic Editor

PLOS ONE

Journal Requirements:

3. Please ensure that you have specified (1) whether consent was informed and (2) what type you obtained (for instance, written or verbal, and if verbal, how it was documented and witnessed). If your study included minors, state whether you obtained consent from parents or guardians. If the need for consent was waived by the ethics committee, please include this information.

This work was supported by departmental funding (Department of Anesthesiology, Universidad del Cauca, Colombia, and Department of Anesthesiology, Erasmus University Medical Centre Rotterdam, The Netherlands).

Reviewers' comments:

Reviewer's Responses to Questions

**Comments to the Author**

1. Is the manuscript technically sound, and do the data support the conclusions?

Reviewer #1: Yes

Reviewer #2: Yes

2. Has the statistical analysis been performed appropriately and rigorously? 

Reviewer #1: N/A

Reviewer #2: Yes

3. Have the authors made all data underlying the findings in their manuscript fully available?

Reviewer #1: Yes

Reviewer #2: Yes

4. Is the manuscript presented in an intelligible fashion and written in standard English?

Reviewer #1: Yes

Reviewer #2: Yes

5. Review Comments to the Author

Reviewer #1: Title: methodological transparency of preoperative clinical practice guidelines for elective surgery. Systematic review.

This is a timely and very important study. Congratulations to all authors.

Please note the following suggestions to further improve the quality of the manuscript:

Line 29: need to be corrected as “..increase quality while decreasing the risk of…”

Line 138: need to be corrected as “All evaluators underwent a thorough detailed training…”

Line 272: need to be corrected as “…interest were declared..”

Line 273: however, it is better to report the proportion of CPG that reported the funding source here.

Line 276: score of 67% cannot be considered as a ‘high score’

Line 276: need to be corrected as “…(67%), with higher scores for CPG..”

Reviewer #2: The review is well described and I have few comments/suggestions.

- In the “search strategy” item, the authors cite the "TRIP Database", but in the "Data sources" item of the abstract it is not mentioned;

- The authors don´t mention when the search in databases was conducted;

- Could you provide list of excluded texts in the supplement (second elegibility)? Could you provide table with the characteristics of the guidelines in the supplement?

- Line 185: "...only six (20%) reported a formal method for achieving..." – the % should be 30% (6 of 20);

-Table 1, in the part where it presents data on clear reporting of funding reports only from 10 guidelines, could it not include the option of "not mentioned" or "not clearly mentioned"?

- In the discussion (lines 245/246): "...obtained AGREE-II scores above 90%;", I believe it should be equal or above 90%, since one of the guidelines presented a score of 90%;

- Domain 6 had a median of 97%, but there were guidelines with low scores (0) in this domain, and it bothered me not to comment on this variability;

- Couldn't a "limitation" be to have considered versions of the same guideline? And language?

- In the results topic, I was bothered by the fact that he reported that 14 guidelines were classified as recommended and mentioned table 1, which despite presenting separate data according to quality classification, is more focused on characteristics, while the evaluation result/scores are in table 2;

- It is missing an arrow in the flowchart (Figure 1);

6. PLOS authors have the option to publish the peer review history of their article (what does this mean?). If published, this will include your full peer review and any attached files.

Reviewer #1: **Yes: **Ishanka Ayeshwari Talagala

Reviewer #2: No

---

## [Author Response · Author response to Decision Letter 0]

3 Dec 2022

We aimed to solve all comments of the reviewers and follow all suggestions to improve the quality and transparency of our paper. We carefully considered all reviewers’ comments, and we are submitting a detailed list of responses (enclosed file) to each comment with quotes from the text, a manuscript with track changes and a revised manuscript.

---

## [Decision Letter · Decision Letter 1]

5 Feb 2023

METHODOLOGICAL TRANSPARENCY OF PREOPERATIVE CLINICAL PRACTICE GUIDELINES FOR ELECTIVE SURGERY. SYSTEMATIC REVIEW.

PONE-D-22-20952R1

Dear Dr. Jose Andres Calvache,

We’re pleased to inform you that your manuscript has been judged scientifically suitable for publication and will be formally accepted for publication once it meets all outstanding technical requirements.

Kind regards,

Seung-Hwa Lee

Academic Editor

PLOS ONE

Additional Editor Comments (optional):

Reviewers' comments:

Reviewer's Responses to Questions

**Comments to the Author**

1. If the authors have adequately addressed your comments raised in a previous round of review and you feel that this manuscript is now acceptable for publication, you may indicate that here to bypass the “Comments to the Author” section, enter your conflict of interest statement in the “Confidential to Editor” section, and submit your "Accept" recommendation.

Reviewer #1: All comments have been addressed

2. Is the manuscript technically sound, and do the data support the conclusions?

Reviewer #1: Yes

3. Has the statistical analysis been performed appropriately and rigorously? 

Reviewer #1: Yes

4. Have the authors made all data underlying the findings in their manuscript fully available?

Reviewer #1: Yes

5. Is the manuscript presented in an intelligible fashion and written in standard English?

Reviewer #1: Yes

6. Review Comments to the Author

Reviewer #1: Congratulations to the authors for the manuscript on a timely and important topic.

Thank you for accepting the suggestions given and making the changes for the manuscript text

7. PLOS authors have the option to publish the peer review history of their article (what does this mean?). If published, this will include your full peer review and any attached files.

Reviewer #1: No

---

## [Editor Report · Acceptance letter]

14 Feb 2023

PONE-D-22-20952R1 

Methodological transparency of preoperative clinical practice guidelines for elective surgery. Systematic Review. 

Dear Dr. Calvache:

I'm pleased to inform you that your manuscript has been deemed suitable for publication in PLOS ONE. Congratulations! Your manuscript is now with our production department. 

Kind regards, 

on behalf of

Dr. Seung-Hwa Lee 

Academic Editor

PLOS ONE